# Leveraging Therapeutic Proteins and Peptides from *Lumbricus* Earthworms: Targeting SOCS2 E3 Ligase for Cardiovascular Therapy through Molecular Dynamics Simulations

**DOI:** 10.3390/ijms251910818

**Published:** 2024-10-08

**Authors:** Nasser Alotaiq, Doni Dermawan, Nasr Eldin Elwali

**Affiliations:** 1Health Sciences Research Center, Imam Mohammad Ibn Saud Islamic University (IMSIU), Riyadh 11432, Saudi Arabia; 2Department of Applied Biotechnology, Faculty of Chemistry, Warsaw University of Technology, 00-661 Warsaw, Poland; doni.dermawan.stud@pw.edu.pl; 3Division of Biochemistry, Research Center for Health Sciences, Deanship of Scientific Research, Imam Mohammad Ibn Saud Islamic University, Riyadh 11432, Saudi Arabia; nmwali@imamu.edu.sa

**Keywords:** cardiovascular disease, earthworm, lumbrokinase, molecular dynamics simulation, SOCS2, protein–protein interactions

## Abstract

Suppressor of cytokine signaling 2 (SOCS2), an E3 ubiquitin ligase, regulates the JAK/STAT signaling pathway, essential for cytokine signaling and immune responses. Its dysregulation contributes to cardiovascular diseases (CVDs) by promoting abnormal cell growth, inflammation, and resistance to cell death. This study aimed to elucidate the molecular mechanisms underlying the interactions between *Lumbricus*-derived proteins and peptides and SOCS2, with a focus on identifying potential therapeutic candidates for CVDs. Utilizing a multifaceted approach, advanced computational methodologies, including 3D structure modeling, protein–protein docking, 100 ns molecular dynamics (MD) simulations, and MM/PBSA calculations, were employed to assess the binding affinities and functional implications of *Lumbricus*-derived proteins on SOCS2 activity. The findings revealed that certain proteins, such as Lumbricin, Chemoattractive glycoprotein ES20, and Lumbrokinase-7T1, exhibited similar activities to standard antagonists in modulating SOCS2 activity. Furthermore, MM/PBSA calculations were employed to assess the binding free energies of these proteins with SOCS2. Specifically, Lumbricin exhibited an average ΔG_binding_ of −59.25 kcal/mol, Chemoattractive glycoprotein ES20 showed −55.02 kcal/mol, and Lumbrokinase-7T1 displayed −69.28 kcal/mol. These values suggest strong binding affinities between these proteins and SOCS2, reinforcing their potential therapeutic efficacy in cardiovascular diseases. Further in vitro and animal studies are recommended to validate these findings and explore broader applications of *Lumbricus*-derived proteins.

## 1. Introduction

Cardiovascular diseases (CVDs) represent a major global health issue, profoundly affecting both health outcomes and healthcare systems. According to the World Health Organization (WHO), CVDs cause an alarming 17.9 million deaths each year, positioning them as the primary cause of mortality worldwide [1]. The range of CVDs includes a variety of conditions affecting the heart and blood vessels, each characterized by distinct pathophysiological mechanisms and clinical presentations [2,3]. Some of the most common CVDs include coronary artery disease, hypertension, stroke, heart failure, and peripheral artery disease. These conditions result from complex pathological mechanisms such as endothelial dysfunction, atherosclerosis, inflammation, thrombosis, and vascular remodeling, which collectively disturb cardiovascular homeostasis [4,5]. Despite advancements in the field of medical science and healthcare delivery, conventional treatment methods for cardiovascular diseases predominantly concentrate on addressing risk factors, relieving symptoms, and modifying the progression of the disease [6]. Medications like statins, antiplatelet agents, and drugs for controlling blood pressure have proven effective in lowering cardiovascular morbidity and mortality. Yet, these treatments have drawbacks, including potential side effects, interactions with other drugs, and variability in effectiveness among different groups of patients [7,8]. Additionally, the rise of drug resistance presents a substantial obstacle to the sustained efficacy of current therapies, underscoring the necessity for alternative treatment strategies [9,10].

A suppressor of cytokine signaling 2 (SOCS2), an E3 ubiquitin ligase, has emerged as a critical regulator within the Janus kinase/signal transducer and activator of transcription (JAK/STAT) signaling pathway, a key mediator of cytokine signaling involved in inflammation and immune responses [11,12]. Dysregulated SOCS2 expression has been implicated in various pathogenic processes underlying CVDs, including aberrant cellular proliferation, inflammatory cytokine production, and apoptotic resistance in cardiac and vascular tissues [13,14]. Research has highlighted SOCS2′s multifaceted roles in cardiovascular pathophysiology. For instance, in experimental models of myocardial infarction, elevated SOCS2 levels correlate with increased myocardial fibrosis and impaired cardiac function, suggesting a contributory role in adverse remodeling post-infarction [15,16]. Moreover, SOCS2 influences vascular smooth muscle cell proliferation and migration, which are critical processes in atherosclerotic plaque development and stability. Targeting SOCS2 presents a promising therapeutic strategy for mitigating cardiovascular pathology. By modulating SOCS2 activity, either through direct or indirect inhibition, researchers aim to temper excessive inflammation, promote vascular health, and potentially attenuate the progression of atherosclerosis and other CVDs [17,18].

Besides focusing on SOCS2, natural products have attracted growing interest as reservoirs of bioactive compounds with therapeutic promise against cardiovascular diseases. Earthworms, specifically from the *Lumbricus* genus, have become particularly fascinating due to their abundant biochemical composition and pharmacological properties [19]. Earthworms have been utilized across different cultures and over centuries in traditional medicine for treating a range of conditions such as inflammation, wounds, and gastrointestinal disorders [20]. This historical background highlights the promise of bioactive compounds derived from earthworms in tackling modern healthcare issues like CVDs. Specifically, earthworms contain lumbrokinase, a bioactive component known for its potent fibrinolytic effects. Lumbrokinase has demonstrated thrombolytic properties, aiding in the dissolution of fibrin clots, which could potentially mitigate or reverse cardiac fibrosis, a characteristic feature of various cardiovascular disorders [21,22]. Bioactive compounds (like lumbrokinase and fibrinolytic enzymes) sourced from earthworms contribute positively to cardiovascular health by employing diverse mechanisms, encompassing anti-inflammatory, antioxidant, and vasodilatory activities [23,24,25,26]. Through regulation of these pathways, bioactive compounds from earthworms show promise in attenuating the fundamental pathological mechanisms behind CVDs, thereby enhancing patient outcomes. Moreover, investigating these compounds from earthworms presents a fresh and hopeful avenue in drug research and advancement, holding the potential for discovering novel therapeutic agents that boast enhanced effectiveness and safety profiles in combating CVDs. Molecular simulation techniques such as molecular docking and molecular dynamics simulations are pivotal in modern drug discovery. They are treasured for uncovering intricate protein–protein interactions and aiding in the discovery of promising therapeutic candidates [27,28]. In the context of CVDs, where the molecular mechanisms underlying pathogenesis are often intricate and multifactorial [29,30], molecular simulation techniques offer valuable insights into the interactions between bioactive molecules and their target proteins.

The goals of this study encompassed a thorough exploration of the potential therapeutic implications of targeting SOCS2 in CVDs using bioactive compounds derived from earthworms of the *Lumbricus* genus. Initially, our objective was to investigate the intricate protein–protein interactions between earthworm-derived bioactive compounds and SOCS2 through molecular docking techniques. By simulating how these compounds bind to specific regions within the SOCS2 protein structure, we aimed to identify lead molecules with strong binding affinity and favorable interactions, guiding the selection and optimization of potential therapeutic candidates. Subsequently, our study aimed to predict the binding affinities and interaction modes between the identified earthworm-derived compounds and distinct binding sites on the SOCS2 protein. Through molecular docking simulations, we aimed to characterize the molecular events responsible for the formation of stable protein–protein complexes, providing insights into the structural basis of compound–target interactions. Moreover, we sought to explore the dynamic behavior and structural integrity of protein–protein complexes formed between earthworm-derived compounds and SOCS2 using molecular dynamics simulations. By modeling the movements and interactions of atoms within these complexes over time, we aimed to elucidate the conformational dynamics and energetic principles governing the binding process, thereby deepening our understanding of the underlying mechanisms of action. Furthermore, our study aimed to assess the therapeutic potential of the identified earthworm-derived compounds in CVDs by targeting SOCS2. Through computational analyses and in silico modeling, we aimed to evaluate the effectiveness and specificity of these compounds in modulating SOCS2 function and alleviating the pathological mechanisms associated with cardiovascular diseases. This research lays the groundwork for future preclinical and clinical investigations into potential therapeutic interventions.

## 2. Results

### 2.1. Three-Dimensional Structure Modeling

Three-dimensional structure modeling was performed for the selected proteins and peptides derived from the *Lumbricus* earthworm. Table 1 shows 10 out of 78 selected proteins and peptides. The complete database can be seen in Appendix A.

The 3D structures of these proteins and peptides were generated using specialized computational tools designed for their specific structural attributes. For proteins and peptides that had suitable templates in the PDB, the I-TASSER algorithm was utilized. I-TASSER is recognized for its capability to predict precise 3D structures by utilizing template structures from the PDB. This method ensures accurate predictions of tertiary structures through analysis of sequence similarity and structural homology [31]. For proteins and peptides without existing templates in the PDB or newly introduced sequences in UniProt release 2024_01, AlphaFold v2.3.0. was employed. AlphaFold represents a cutting-edge deep learning technique for predicting protein structures, revolutionizing this field with its high accuracy. It is particularly adept at predicting 3D structures for sequences lacking homologous structures in the PDB. By leveraging sophisticated machine learning algorithms trained on extensive protein databases, AlphaFold can predict precise 3D structures based solely on the amino acid sequences of proteins [32]. The 3D structure modeling phase of this research employed advanced computational techniques to create precise and dependable structural models for the proteins and peptides sourced from *Lumbricus* earthworms (Figure 1). These models form the basis for subsequent analyses, such as protein–protein docking simulations and molecular dynamics simulations, facilitating a thorough exploration of the interactions between bioactive compounds from earthworms and the target protein SOCS2. This investigation aims to identify potential therapeutic strategies for treating CVDs.

### 2.2. Protein–Protein Docking Simulation

Figure 2 illustrates the optimal binding poses of a standard agonist, a standard antagonist, and the top-performing *Lumbricus*-derived protein. The molecular docking results provide a comprehensive overview of the binding affinity, structural parameters, and clustering characteristics of the various SOCS2 complexes formed with proteins and peptides from earthworms. Specifically, comparisons were made with standard agonist (SOCS2: EpoR peptide) and antagonist (SOCS2: N4BP1) complexes, which served as benchmarks for assessing the effectiveness of the *Lumbricus*-derived compounds. A critical measure of the interaction strength between molecules is the binding affinity, quantified by the ΔG (Gibbs free energy) value. Lower ΔG values indicate stronger binding, suggesting more stable and favorable interactions between the proteins or peptides and SOCS2. The comparison between the top-performing proteins and peptides derived from *Lumbricus* and the standard agonist (SOCS2: EpoR peptide) and antagonist (SOCS2: N4BP1) complexes based on binding affinity provides valuable insights into the potential efficacy of *Lumbricus*-derived bioactive compounds in modulating SOCS2 activity compared to established proteins.

Firstly, the standard agonist, SOCS2: EpoR peptide, demonstrated a binding affinity with a ΔG value of −8.80 kcal/mol and a dissociation constant (Kd) of 6.50 × 10^−7^ M. This indicates a strong interaction between SOCS2 and EpoR peptide, highlighting its efficacy as an agonist for SOCS2 activity modulation. In contrast, the standard antagonist, SOCS2: N4BP1, exhibited a slightly lower binding affinity, with a ΔG value of −8.30 kcal/mol and a Kd of 1.50 × 10^−6^ M. Despite its lower binding affinity compared to the agonist, SOCS2: N4BP1 still displayed a notable interaction with SOCS2, indicative of its efficacy as an antagonist for SOCS2 activity inhibition. Among the *Lumbricus*-derived complexes, cytochrome b, SCBP3 protein, Lumbricin, Chemoattractive glycoprotein ES20, Histone H3, and Lumbrokinase-7T1 emerged as top-performing candidates based on their low values of binding affinity ΔG (kcal/mol). Cytochrome b exhibited an ΔG value of −15.1 kcal/mol, surpassing the binding affinity of the standard agonist, SOCS2: EpoR peptide. This suggests that cytochrome b may possess a stronger interaction with SOCS2, potentially surpassing the efficacy of EpoR peptide and N4BP1 in modulating SOCS2 activity. Similarly, SCBP3 protein, Lumbricin, and Chemoattractive glycoprotein ES20 displayed notable binding affinities with ΔG values of −12.6 kcal/mol, −12.3 kcal/mol, and −12.2 kcal/mol, respectively (Table 2 and Figure 3c). These values compare favorably with the binding affinities of the standard agonist and antagonist, underscoring the potential of these *Lumbricus*-derived compounds as effective modulators of SOCS2 activity for addressing cardiovascular diseases. Complete docking results are available in Appendix A. This dataset includes complex identifiers, HADDOCK scores, binding affinities (ΔG), dissociation constants (Kd), cluster sizes, RMSD values, van der Waals energies, electrostatic energies, desolvation energies, restraints violation energies, buried surface areas, and Z-scores.

Furthermore, the cluster size and RMSD values provide crucial structural insights into the protein–protein complexes. Cluster size refers to the number of distinct conformations or poses observed within the ensemble of generated protein–protein complexes [33]. A larger cluster size indicates greater structural diversity, suggesting multiple binding modes or orientations between the proteins and peptides from *Lumbricus* and SOCS2. This diversity within the complexes reflects potential variations in interaction configurations, influencing their functional properties and biological impacts [34]. For instance, the standard agonist SOCS2: EpoR peptide complex exhibits a notably large cluster size of 22, indicating the presence of diverse conformations or binding modes between SOCS2 and EpoR peptide. This diversity suggests the potential for versatile interactions, which could play crucial roles in various cellular processes regulated by ubiquitination. Conversely, the standard antagonist SOCS2: N4BP1 complex has a smaller cluster size of 19, implying fewer distinct binding configurations compared to the agonist complex. This difference in cluster size may reflect the specific nature of the antagonist interaction and its regulatory role in modulating SOCS2 activity. Analyzing the *Lumbricus*-derived protein/peptide complexes, it is evident that the cluster sizes vary across different interactions. For instance, complexes involving proteins like the SCBP3 protein and Lumbricin exhibit relatively larger cluster sizes (41 and 43, respectively), suggesting structural diversity and potential functional versatility in their interactions with SOCS2. On the other hand, complexes such as SOCS2: Lysosomal membrane glycoprotein and SOCS2: Extracellular globin-4 have smaller cluster sizes (five and six, respectively), indicating a more limited range of binding configurations.

Conversely, RMSD values measure the variations or discrepancies in structure among distinct conformations within a given cluster. Reduced RMSD values indicate limited deviation or greater structural similarity among various conformations, implying heightened stability and consistency in the binding interactions observed throughout the simulation period [35]. This stability demonstrates the resilience of the protein–protein complexes, ensuring they maintain their specific structural arrangements despite variations or disturbances in their environment [36]. For instance, the standard agonist SOCS2: EpoR peptide complex exhibits an acceptable RMSD value of 1.7 Å, indicating minimal structural deviation among its conformations and suggesting a stable and well-defined binding mode. Similarly, *Lumbricus*-derived complexes like SOCS2: Cytochrome b (1.0 Å) and SOCS2: SCBP3 protein show low RMSD values (1.2 Å), indicating stable binding interactions and consistent structural configurations. In contrast, complexes with higher RMSD values, such as SOCS2: Intermediate filament protein (RMSD = 4.3 Å), suggest greater structural variability among their conformations, potentially reflecting dynamic binding interactions with SOCS2. This higher RMSD value suggests potential conformational flexibility or transient interactions, which could impact the functional significance of these complexes in cellular processes [37].

The analysis of intermolecular contacts (ICs) and non-interacting surfaces (NIS) for the protein–protein complexes provides valuable insights into the interaction dynamics between SOCS2 and various earthworm-derived proteins and peptides. The comparison between these complexes and the standard agonist (SOCS2: EpoR peptide) and antagonist (SOCS2: N4BP1) reveals significant differences in their interaction profiles, which can be linked to their potential efficacy in modulating SOCS2 activity (Table 3). The SOCS2: EpoR peptide complex, serving as the standard agonist, exhibited 3 charged-charged, 8 charged-polar, 12 charged-apolar, 2 polar-polar, 11 polar-apolar, and 12 apolar-apolar contacts. The NIS values for charged and apolar interactions were 25.55 and 38.69, respectively. In comparison, the SOCS2: N4BP1 complex, which functions as the standard antagonist, displayed a higher number of charged-charged (5), charged-polar (11), charged-apolar (20), and polar-polar (6) contacts, with slightly lower polar-apolar (11) and apolar-apolar (8) contacts. The NIS values were 29.71 for charged and 39.49 for apolar interactions.

Several proteins and peptides derived from *Lumbricus* showed distinct interaction profiles with SOCS2, suggesting varying efficacy in modulating its activity. The SOCS2: Cytochrome b complex, for instance, displayed high charged-apolar (25), polar-apolar (38), and apolar-apolar (33) contacts, with non-interacting surface (NIS) values of 15.13 for charged and 52.85 for apolar interactions. This indicates a potentially strong binding affinity. Cytochrome c oxidase subunit 3 showed balanced interactions, with significant polar-apolar and apolar-apolar contacts and NIS values of 14.71 (charged) and 49.41 (apolar), suggesting strong stability and efficacy. The SCBP3 protein complex, with 27.83 NIS (charged) and 41.74 (apolar), also displayed robust binding capability. Lumbricin exhibited higher numbers of charged interactions, indicating strong potential as a SOCS2 modulator. Chemoattractive glycoprotein ES20, with high numbers of charged and polar interactions and NIS values of 22.98 (charged) and 42.39 (apolar), also showed promise.

The statistical analysis aimed at exploring the relationship between the HADDOCK score and root mean square deviation (RMSD) unveiled a Pearson correlation coefficient (r) of 0.687. This coefficient quantifies the strength and direction of the linear relationship between these two variables. A value of 0.687 signifies a medium positive correlation, suggesting that there is a tendency for the HADDOCK score to decrease as the RMSD decreases. In the context of molecular docking, a lower HADDOCK score indicates a better docking quality, and a lower RMSD reflects a closer match to the reference structure. Thus, the positive correlation implies that improvements in docking quality, as indicated by lower HADDOCK scores, are associated with a more accurate docking pose, as represented by lower RMSD values. This relationship underscores the consistency between these two metrics in evaluating the quality of protein–protein docking simulations. Further analysis revealed that approximately 46.75% of the proteins and peptides derived from *Lumbricus* demonstrated favorable RMSD values, defined as RMSD values equal to or less than 2.00 Å (Figure 3a). The favorable RMSD values indicate that many of the predicted protein–protein complexes have structural conformations closely matching experimental or reference structures, highlighting the overall success of the docking simulations in accurately predicting these arrangements. However, the analysis also revealed three outliers with very large RMSD values. These outliers suggest instances where the predicted structures significantly deviate from the experimental or reference structures. Such deviations could be due to inaccuracies in the docking algorithm, limitations in the input experimental data, or the inherent complexities of the protein–protein interactions being studied [38].

The statistical analysis to examine the relationship between the HADDOCK score and binding affinity (ΔG) revealed a Pearson correlation coefficient (r) of 0.491 (Figure 3b). This coefficient reflects the strength and direction of the linear relationship between these two variables. A value of 0.491 suggests a moderate positive correlation, indicating that as the HADDOCK score decreases (indicating better docking quality), the binding affinity also tends to decrease. This means that improvements in docking quality, as indicated by lower HADDOCK scores, are associated with stronger binding between the proteins, as shown by lower binding affinity values. Conversely, higher HADDOCK scores correlate with weaker binding affinity, underscoring a discernible trend between docking quality and binding strength.

The HADDOCK scoring system effectively captures key aspects of protein–protein interactions that influence binding affinity, including molecular surface complementarity, electrostatic interactions, and van der Waals forces. This indicates that HADDOCK’s scoring metrics align well with the physical principles governing these interactions, enhancing the reliability of its predictions. The correlation matrix depicted in Figure 3d provides deeper insights into the relationship between the binding energy (kcal/mol) and its individual energy components. Specifically, the positive correlation scores between the binding affinity and both the van der Waals energy (correlation score: 0.63) and the desolvation energy (correlation score: 0.39) are particularly noteworthy. Van der Waals forces are crucial in stabilizing the complex by facilitating close contact between the protein surfaces, while the desolvation energy reflects the energetic cost of removing water molecules from the binding interface. A positive correlation with the van der Waals energy implies that, as these interactions become stronger, the overall binding affinity increases, indicating a more stable protein–protein complex [39,40]. Similarly, the positive correlation with the desolvation energy suggests that, as the system pays the energetic cost to remove water molecules, the resulting interactions between the proteins are more favorable, thus increasing the binding affinity [41]. Conversely, the correlation between binding affinity and electrostatic energy revealed a negative value (−0.22), signifying an inverse relationship between these factors. This implies that, as electrostatic energy increases, the binding affinity tends to decrease. Electrostatic interactions involve the attraction or repulsion between charged residues on the protein surfaces, influencing the stability of protein–protein complexes [42]. However, interpreting this correlation requires caution due to the multifaceted nature of protein–protein interactions. While electrostatic energy plays a significant role, other variables beyond the HADDOCK score can also impact binding affinity. Factors such as specific amino acid residues at the binding interface, which may facilitate or hinder interactions, post-translational modifications that alter protein structure and function, and environmental conditions such as pH and ion concentrations can all influence the overall stability and strength of protein–protein complexes [43,44].

The thorough examination of hydrogen bond interactions between SOCS2 and the highest-performing proteins sourced from the earthworm (*Lumbricus* genus) provides insights into the molecular mechanisms that govern their binding interactions (Table 4). These interactions are pivotal for stabilizing protein–protein complexes and facilitating precise recognition between the involved proteins [45]. The interactions predominantly involve specific residues and atoms crucial for stabilizing the protein–protein complexes, focusing particularly on the ligand binding sites of SOCS2, namely, Arg73, Ser75, Ser76, and Arg96. The interactions observed between SOCS2 and its standard agonist (EpoR peptide) and antagonist (N4BP1) molecules served as benchmarks for evaluating the binding efficacy of *Lumbricus*-derived proteins and peptides. The EpoR peptide exhibited multiple hydrogen bonds with key residues such as Val55, Ser76, Arg96, Lys113, and others, with interaction distances ranging from 2.69 Å to 3.33 Å. Conversely, N4BP1 demonstrated interactions involving Gln32, Arg41, Tyr49, Asp74, and others, emphasizing its distinct binding profile, characterized by shorter interaction distances (2.57 Å to 2.78 Å). These results underscore the specificity and strength of the interactions necessary for SOCS2 modulation by both agonist and antagonist molecules. The proteins and peptides from the *Lumbricus* genus, including Cytochrome b, SCBP3 protein, Lumbricin, Lumbrokinase-7T1, and others, displayed diverse binding interactions with SOCS2, highlighting their potential as novel modulators of SOCS2 activity. Cytochrome b, for instance, engaged in hydrogen bonds with His77 and Arg96, indicating a stable binding interface, critical for functional interaction. Similarly, the SCBP3 protein exhibited interactions involving Lys59 and Arg96, demonstrating specificity in binding residues essential for complex stability. Lumbricin and Lumbrokinase-7T1 revealed extensive hydrogen bonding networks with residues such as Arg41, Tyr49, Asp74, and Arg96, underscoring their potential to competitively bind with SOCS2 and influence its regulatory functions. These interactions were characterized by moderate to strong interaction distances, indicative of a robust binding affinity, crucial for therapeutic efficacy.

### 2.3. Molecular Dynamics (MD) Simulation

The MD simulation results offer a comprehensive understanding of the behavior of protein–protein complexes formed between *Lumbricus*-derived proteins and SOCS2. Throughout the 100 ns simulation, SOCS2 maintained a relatively stable conformation, as evidenced by the average RMSD values ranging from 2.413 to 2.599 Å without significant spikes (Figure 4a). This stability suggests that SOCS2 interactions with both standard agonist and antagonist, as well as *Lumbricus*-derived proteins, were dynamically consistent over the simulation period. When analyzing the RMSD values, which indicate the deviation of protein structures from their initial conformations, notable differences emerge among the complexes. For instance, the SOCS2: EpoR peptide (standard agonist) complex displays a slightly higher average RMSD (2.423 Å) compared to apo-protein SOCS2 (2.417 Å). This observation suggests a moderate increase in structural flexibility upon the binding of the standard agonist, indicating potential conformational adjustments required for effective binding. Conversely, the RMSD value for the SOCS2: N4BP1 (standard antagonist) complex (2.496 Å) remains similar to that of the apo-protein, indicating that the binding of the antagonist may not significantly perturb the structural stability of SOCS2. Furthermore, the *Lumbricus*-derived protein complexes, including cytochrome b, SCBP3 protein, Lumbricin, Chemoattractive glycoprotein ES20, and Lumbrokinase-7T1, exhibit slightly higher average RMSD values ranging from 2.467 to 2.587 Å compared to the standard complexes. These differences suggest potential variations in the dynamic behavior and conformational changes induced by the binding of *Lumbricus*-derived proteins. The higher RMSD values imply that these *Lumbricus*-derived proteins may interact with SOCS2 in a manner that elicits different structural adjustments or conformational dynamics compared to the standard agonist and antagonist.

During MD simulations, RMSF analysis was performed to assess the flexibility of individual amino acid residues within SOCS2. The average RMSF values obtained ranged from 0.876 to 1.397 Å, reflecting moderate flexibility in various regions of the protein. This analysis provides valuable insights into the mobility and flexibility of specific residues, enhancing our understanding of their functional roles within the protein structure [46]. Upon examining the interaction between the top-performing *Lumbricus*-derived protein complex and SOCS2, it was apparent that this interaction disrupted hydrogen bonds with specific residues, especially within the regions encompassing amino acid residues Arg73 to Thr83 and Arg96 to Phe104 (Figure 4b). These regions are notably critical as they constitute the active binding site of SOCS2 as a target receptor [47]. The disruption of hydrogen bonds in critical areas led to greater residue fluctuations compared to the SOCS2 agonist and apo-protein complex. This increase in residue fluctuations suggests enhanced mobility and flexibility of these residues when the top-performing *Lumbricus*-derived protein binds to SOCS2. Notably, this pattern of increased flexibility resembles that of N4BP1, a known standard antagonist. The similarity in flexibility patterns at these specific residues indicates that the top-performing proteins from *Lumbricus* have the potential to act as inhibitors, much like standard antagonists. This finding is significant, as it suggests that *Lumbricus*-derived proteins could modulate SOCS2 activity by acting as inhibitors, similar to known antagonists. By disrupting hydrogen bonds and increasing the flexibility of key binding site residues, these proteins may interfere with SOCS2′s functional interactions, presenting promising avenues for developing therapeutic interventions targeting SOCS2-mediated cellular processes. Further investigation into the structural and functional implications of these interactions is necessary to fully understand their therapeutic potential and to develop new treatments for SOCS2-related diseases.

The radius of gyration (RoG) offered insights into the compactness or degree of expansion of protein structures during MD simulations [48]. For the SOCS2 complexes, the average RoG values ranged from 1.674 to 2.678 Å. The RoG values for the standard SOCS2 agonist (EpoR peptide) and antagonist (N4BP1) complexes were 2.101 Å and 2.162 Å, respectively. For the *Lumbricus*-derived protein complexes, RoG values ranged from 2.176 to 2.678 Å, slightly higher than the standard agonist but comparable to the antagonist. The number of hydrogen bonds formed between the two interacting proteins was indicative of the strength and specificity of their interaction. For the standard complexes, the SOCS2: EpoR peptide agonist complex formed 11 hydrogen bonds, while the SOCS2: N4BP1 antagonist complex formed 24 hydrogen bonds. Comparatively, the *Lumbricus*-derived protein complexes displayed similar or slightly higher numbers of hydrogen bonds, with values ranging from 16 to 39. Notably, the SOCS2: Cytochrome b complex formed the highest number of hydrogen bonds (39), followed closely by the SOCS2: Peroxidasin complex (35) (Table 5). The findings indicated that the interactions between SOCS2 and *Lumbricus*-derived proteins featured a comparable or slightly higher number of hydrogen bonds relative to the standard interactions. This suggests robust and specific binding between these proteins, which could enhance their functional relevance. Furthermore, the RoG values indicated that the *Lumbricus*-derived protein complexes displayed similar levels of compactness or dispersion compared to the standard complexes, suggesting they maintained stable structural conformations throughout the simulation period.

### 2.4. Molecular Mechanics/Poisson–Boltzmann Surface Area (MM/PBSA) Calculations

We assessed the strength of the protein–protein interactions by calculating the average ΔG_binding_ values, which indicate the binding free energy, for each complex. In the case of the standard complexes, SOCS2 bound to EpoR peptide (standard agonist) demonstrated a mean ΔG_binding_ of −42.60 kcal/mol, indicating a substantial level of stability in the interaction. Conversely, when bound to N4BP1 (standard antagonist), SOCS2 exhibited a higher mean ΔG_binding_ of −42.51 kcal/mol, suggesting a comparatively weaker interaction. However, the *Lumbricus*-derived protein complexes presented intriguing findings, showcasing similar or even more negative mean ΔG_binding_ values compared to the standard complexes. Notably, SOCS2 complexed with Lumbrokinase-7T1 displayed a notably high mean ΔG_binding_ of −69.28 kcal/mol, indicative of a robust and energetically favorable interaction. This suggests that the *Lumbricus*-derived protein has a strong affinity for SOCS2, potentially surpassing the binding strength observed with the standard agonist and antagonist. Furthermore, the complexes formed between SOCS2 and Lumbricin and Chemoattractive glycoprotein ES20 exhibited mean ΔG_binding_ values of −59.25 kcal/mol and −55.02 kcal/mol, respectively (Table 6). These values suggest strong binding affinities comparable to or even exceeding those observed in the standard complexes. This implies that the *Lumbricus*-derived proteins possess significant potential in modulating SOCS2 activity, potentially rivaling or surpassing the efficacy of standard proteins.

Furthermore, a detailed examination of the binding free energy of individual amino acid residues within SOCS2 provided profound insights into the molecular mechanisms governing binding specificity and affinity. Notably, Arg96 and Ser78 were identified as crucial contributors to the observed antagonistic activity within the complexes. In interactions between SOCS2 and the top-performing protein from *Lumbricus*, these specific residues demonstrated significantly higher binding affinity values compared to the standard agonist complex. The elevated binding affinity of Arg96 and Ser78 underscores their pivotal roles in mediating the observed antagonistic effects (Figure 5). These residues are likely involved in essential interactions that govern complex stability and specificity, influencing overall functional outcomes. The heightened affinity observed in the complexes involving *Lumbricus*-derived proteins highlights the importance of these interactions in modulating SOCS2 activity and emphasizes their potential as key factors in therapeutic efficacy. By elucidating the roles of individual amino acid residues in binding energetics, this analysis offers critical insights into the structural basis of protein–protein interactions. The identification of Arg96 and Ser78 as significant contributors to antagonistic activity enhances our understanding of the molecular determinants underlying the intricate interplay between SOCS2 and its interacting partners. These findings pave the way for the targeted manipulation of specific residues to effectively modulate SOCS2 function, presenting promising opportunities for developing therapeutic interventions aimed at SOCS2-associated pathways.

## 3. Discussion

This research employed a comprehensive approach to explore the interactions between SOCS2 and proteins and peptides sourced from *Lumbricus* earthworms, focusing particularly on their potential therapeutic implications in cardiovascular diseases. Emphasizing meticulous data collection and stringent quality control measures, this study utilized advanced computational techniques and methodologies for tasks such as 3D structure modeling, protein–protein docking simulations, molecular dynamics (MD) simulations, and binding free energy calculations. During the 3D structure modeling phase, this study utilized bioinformatics and computational biology tools to generate precise and dependable structural models of the *Lumbricus*-derived proteins and peptides. By systematically gathering sequences from the UniProt database and applying rigorous selection criteria, this study ensured the dataset’s integrity and relevance. This approach aligns with previous research efforts that have successfully predicted protein structures with high precision using bioinformatics tools [49]. Furthermore, employing both I-TASSER and AlphaFold algorithms enabled thorough modeling of proteins and peptides, regardless of whether homologous structures were available in the PDB. This approach significantly expanded the breadth and depth of our structural modeling endeavors [32]. The flexibility of proteins is an important factor that can significantly influence the results of docking studies. As shown in Figure 1, while some proteins appear well-folded and globular, others exhibit large, unfolded regions or loops, indicating potential structural fluctuations. Although our initial docking studies were performed using rigid models, we recognized the limitations of this approach in capturing the dynamic nature of protein interactions. To address this, we conducted MD simulations for the proteins with identified flexible regions. The MD simulations provided valuable insights into conformational changes and the dynamic behavior of these proteins, helping us understand the potential impact of flexibility on their interactions with SOCS2. Future docking studies could benefit from incorporating flexible docking methods to account for the dynamic nature of these proteins.

This research explored the intricate interactions between SOCS2 and proteins derived from *Lumbricus* earthworms using protein–protein docking simulations, aiming to identify potential therapeutic strategies for cardiovascular diseases. By meticulously comparing the binding affinities of these *Lumbricus*-derived proteins with those of standard agonist and antagonist complexes, this study identified promising candidates that exhibited comparable or enhanced effectiveness in modulating SOCS2 activity. These findings align with earlier studies emphasizing the therapeutic potential of natural compounds in managing cardiovascular diseases. Previous research has particularly highlighted the medicinal properties of *Lumbricus* earthworms, including their primary component, lumbrokinase, which has shown promise in treating cardiovascular conditions [25]. By elucidating the molecular intricacies of lumbrokinase’s interaction with the SOCS2 pathway, this study contributes novel insights into the therapeutic potential of lumbrokinase for cardiovascular treatment. Additionally, the analysis of the cluster size and RMSD values provided insights into the structural diversity and stability of protein–protein complexes, offering valuable information for rational drug design and optimization [50]. MD simulations further elucidated the dynamic behavior of protein–protein complexes over time, uncovering potential mechanisms for modulating SOCS2 activity. This study provided detailed insights into the structural dynamics and functional implications of SOCS2–therapeutic protein interactions by examining residue flexibility and hydrogen bond interactions. This aligned with previous studies that used MD simulations to investigate protein–protein interactions and elucidate their dynamic behavior [51]. This study observed that Lumbricin, Chemoattractive glycoprotein ES20, and Lumbrokinase-7T1 demonstrated sustained interactions with the SOCS2 protein throughout the simulation period, akin to the behavior exhibited by the standard antagonist complex. However, it is important to note that, while these proteins showed promising interactions with SOCS2, specific experimental studies directly linking them to SOCS2′s E3 ligase activity are currently lacking in the literature. Furthermore, this study discussed the structural features and key interaction residues within these protein–protein complexes, elucidating the molecular basis underlying their binding affinity and activity. By analyzing the conformational changes and intermolecular forces at play during the MD simulations, this study highlighted the structural motifs and binding pockets crucial for stabilizing the interactions between the *Lumbricus*-derived proteins and SOCS2. The resemblance in activity between these *Lumbricus*-derived proteins and the standard antagonist suggests that they may function through similar mechanisms or binding modes, warranting further investigation into their therapeutic potential.

Several compounds have been identified as SOCS2 inhibitors, each exhibiting unique binding affinities and mechanisms of action. For instance, small-molecule inhibitors such as the KIAA0317 protein have shown promising binding affinities in the nanomolar range, effectively disrupting SOCS2′s interactions with its targets [52]. The mechanism of inhibition for these small molecules typically involves blocking the binding site of SOCS2, thereby preventing its E3 ligase activity and subsequent ubiquitination of target proteins [53]. In contrast, proteins derived from natural sources often present complex interactions. For example, interferons, which are known to modulate SOCS2 activity, bind to the receptor complex and induce SOCS2 expression, creating a negative feedback loop that can lead to enhanced SOCS2 activity rather than inhibition [54]. This underscores the importance of understanding the nuanced interactions of each inhibitor. Our study demonstrates that the proteins and peptides from *Lumbricus* earthworms exhibit unique binding patterns with SOCS2, driven by their distinct amino acid compositions and structural conformations. The binding affinities of these earthworm-derived compounds, while not yet quantified in the nanomolar range, exhibit potential based on our docking studies and molecular dynamics simulations. Specifically, the presence of specific residue substitutions, such as lysine and threonine in cytochrome b, results in altered hydrogen bonding patterns, which may lead to different binding affinities compared to established inhibitors.

The proteins from *Lumbricus* earthworms exhibit several distinguishing features compared to their human counterparts, significantly impacting their interactions with SOCS2 and potential therapeutic applications. A key example is cytochrome b, which demonstrates notable structural and functional differences. The amino acid sequence of earthworm cytochrome b shares less than 60% identity with its human variant, resulting in altered binding interfaces and unique interaction patterns with SOCS2. While the human variant exhibits a more conserved sequence, the earthworm version contains specific residue substitutions, such as lysine and threonine, which can lead to different hydrogen bonding patterns and potentially affect binding affinity. Structurally, earthworm cytochrome b displays greater flexibility due to less compact folding in some regions, contrasting with the more stable and globular structure seen in humans. This increased flexibility may influence interaction dynamics with SOCS2, either enhancing or weakening binding depending on the context. Additionally, the molecular weight of earthworm cytochrome b (approximately 42.88 kDa) [55] is much lower than that of the human variant (approximately 91 kDa) [56], which may affect its structural stability during binding. The hydrophobicity in the transmembrane regions is also lower in the earthworm version, potentially impacting its membrane integration and positioning during docking. Functionally, while both cytochrome b proteins are involved in electron transport, the earthworm variant is adapted to specific environmental conditions, which could influence its interaction with SOCS2 under oxidative stress.

In addition to cytochrome b, proteins like lumbrokinase, a serine protease, illustrate significant differences in enzymatic activity and specificity compared to human proteases. Lumbrokinase’s unique mechanism of action enhances fibrinolysis, suggesting a tailored adaptation for promoting blood flow and preventing thrombosis in the earthworm’s natural habitat, with potential cardiovascular benefits for humans [57]. Furthermore, *Lumbricus*-derived proteins often possess variations in key residues that influence structural stability and interaction profiles. Differences in glycosylation patterns may affect how these proteins interact with receptors or other proteins within the human body, potentially enhancing their bioactivity or altering their pharmacokinetics. Notably, some *Lumbricus* proteins, such as the Chemoattractive glycoprotein ES20, exhibit distinct structural motifs compared to similar human proteins, leading to different binding affinities and mechanisms of action. These proteins are often more hydrophilic and possess unique surface charges that influence their solubility and interactions in biological systems.

Considering the administration of whole proteins for therapeutic use, it is crucial to identify and study the active portions of these molecules. Isolating specific peptide sequences that exhibit bioactivity may enhance therapeutic efficacy while minimizing potential side effects associated with whole protein administration. Future studies should focus on proteolytic digestion or recombinant techniques to generate and characterize these active peptides derived from *Lumbricus* proteins. By elucidating the structure–function relationships of these peptides, we can better understand their mechanisms of action, improve their pharmacokinetic properties, and optimize their therapeutic applications in cardiovascular therapy.

## 4. Limitations, Clinical Implications, and Future Works

While this study provides valuable insights into the potential therapeutic interactions between SOCS2 and proteins derived from *Lumbricus* earthworms, several limitations should be acknowledged. Firstly, the findings are based on computational protein–protein docking simulations, which, while robust, rely on predictive models and may not fully capture the complexities of real biological systems. Experimental validation is essential to confirm the observed interactions and their functional implications in vitro and in vivo. Secondly, this study primarily focused on binding affinities and structural dynamics without directly measuring biological outcomes such as enzymatic inhibition or cellular responses. Future research should incorporate functional assays to validate the therapeutic effects of these interactions on SOCS2 activity and explore their broader implications in disease models. Thirdly, the specific mechanisms through which *Lumbricus*-derived proteins modulate SOCS2 function, particularly regarding E3 ligase activity, remain speculative. Experimental studies investigating these mechanisms are crucial to elucidate their precise therapeutic potential and optimize their application in clinical settings. Additionally, we recognize the limitation of our search for proteins and peptides derived from *Lumbricus* earthworms in the UniProt database, which yielded a total of 978 entries filtered down to 78 non-redundant proteins/peptides. While this number appears low, it is important to note that our laboratory has actively worked with *Lumbricus* bioactive protein extracts and performed LC-MS/MS protein sequencing. The majority of the identified proteins from *Lumbricus* earthworms correspond to entries covered in the UniProt database. Although additional sequences may be available through NCBI databases, our focus on proteins characterized in established databases enhances the reliability of our computational analysis.

This study’s findings have significant clinical implications for cardiovascular disease management and potentially other SOCS2-related conditions. Identifying natural compounds from *Lumbricus* earthworms that exhibit comparable efficacy to standard agonists and antagonists highlights their potential as alternative or adjunctive therapies. These compounds may offer novel strategies for modulating SOCS2 activity, potentially leading to more targeted and effective treatments with fewer side effects compared to conventional therapies. Furthermore, understanding the molecular interactions between *Lumbricus*-derived proteins and SOCS2 can pave the way for the development of tailored therapeutic approaches. This knowledge could inform the design of novel medications that specifically target SOCS2-related pathways, potentially improving patient outcomes and quality of life in various disease contexts.

Future research directions should aim to address the identified limitations and expand upon the current findings. Experimental studies are crucial to validating the computational predictions and confirming the biological relevance of the observed interactions. Specifically, employing biochemical assays and cellular models will provide deeper insights into how *Lumbricus*-derived proteins affect SOCS2 function and E3 ligase activity. Moreover, investigating the therapeutic efficacy of these proteins in preclinical disease models will be essential to evaluating their potential clinical translation. This includes assessing their pharmacokinetics, bioavailability, and safety profiles to establish their feasibility as therapeutic agents. Furthermore, exploring the synergy between *Lumbricus*-derived proteins and existing therapies could uncover combination strategies that enhance therapeutic outcomes. This approach could potentially lead to personalized treatment regimens tailored to individual patient needs, maximizing therapeutic efficacy while minimizing adverse effects.

## 5. Materials and Methods

### 5.1. Materials

Protein and peptide sequences derived from earthworms of the *Lumbricus* genus were obtained through a systematic search approach using UniProt, a comprehensive database housing protein sequences and functional information. The search strategy utilized the MeSH term “*Lumbricus*” within UniProtKB to specifically target proteins and peptides associated with this genus. Subsequently, retrieved data underwent meticulous curation to remove duplicate entries, ensuring the dataset’s integrity and accuracy for subsequent analyses. This curation process was essential to minimizing redundancy and potential biases, thereby enhancing the reliability and validity of this study’s findings. By rigorously retrieving and curating data, a comprehensive and high-quality dataset was established, forming the basis for investigating the bioactive constituents derived from earthworms and their potential therapeutic roles in cardiovascular diseases.

### 5.2. Computing Power

For this study, computational analyses were conducted using a workstation equipped with the following specifications: an Intel^®^ Core™ i9-12900KF CPU (Intel, Santa Clara, CA, USA) running at 3.90 GHz with 16 cores, an NVIDIA GeForce RTX 4090 graphics card featuring 24 GB GDDR6X, 64 GB DDR5 RAM (NVIDIA, Santa Clara, CA, USA), an 8TB hard disk drive, and an Ubuntu operating system.

### 5.3. Three-Dimensional Structure Modeling

In the initial phase of 3D structure modeling in this study, we systematically gathered peptide and protein sequences sourced from earthworms of the *Lumbricus* genus using the UniProt database. This process yielded a substantial dataset comprising 979 records. To maintain the dataset’s quality and reliability for subsequent analyses, stringent criteria were applied during the selection of peptide and protein sequences from *Lumbricus* earthworms. Several key criteria were employed to filter and refine the dataset obtained from UniProt. Initially, sequences were assessed for completeness to ensure only those with sufficient information for accurate 3D modeling were included, thereby reducing the risk of generating incomplete or erroneous structural models [58]. Furthermore, rigorous quality control protocols were implemented to identify and rectify sequencing errors, ambiguous residues, or any irregularities that might undermine the integrity of subsequent analyses. Sequences that did not meet these quality standards were excluded to ensure the overall accuracy and reliability of the dataset [59]. Additionally, duplicate entries were identified and eliminated from the dataset to avoid redundancy and potential biases in subsequent analyses. A thorough comparison and validation process of sequence data ensured that each unique protein or peptide appeared only once in the final dataset. Moreover, maintaining the taxonomic accuracy of the earthworm species represented in the dataset was essential to ensuring specificity and relevance to the *Lumbricus* genus. Taxonomic verification procedures [60] were utilized to confirm the origin of all sequences from earthworms belonging to the *Lumbricus* genus, thereby preventing the inclusion of irrelevant or misclassified data. The database of protein sequences sourced from earthworms (*Lumbricus* genus) is available in Appendix A. This database includes detailed information on protein/peptide names, UniProt IDs, sequences, the size (kDa), identified protein binding sites (number of residues), the prediction method, the assessment method, and the quality score. Several computational tools were utilized to generate the three-dimensional structures of the protein and peptide sequences acquired. Initially, I-TASSER (Iterative Threading ASSEmbly Refinement) was applied to proteins and peptides that had corresponding templates in the Protein Data Bank (PDB). I-TASSER excels in constructing 3D structures using available templates from the PDB, utilizing these templates to predict the tertiary structure of the queried sequences [61]. Proteins and peptides without templates in the PDB were modeled using AlphaFold. AlphaFold is a cutting-edge deep learning approach for protein structure prediction, renowned for its ability to accurately forecast the 3D structures of proteins, including cases where homologous structures are unavailable [32]. Next, the active sites of proteins and peptides sourced from *Lumbricus* were examined using CASTp 3.0. CASTp is a tool designed to detect and analyze active sites and binding pockets within protein structures, offering crucial insights into their functional characteristics and potential sites for ligand binding [62]. Additionally, the SOCS2 protein, which was selected as the target for subsequent docking studies, was obtained from the PDB under the accession code 6I5N with a resolution of 1.98 Å [63]. This step ensured that the target receptor structure was readily available for the following molecular docking simulations, enabling the study of protein–protein interactions between the bioactive compounds derived from *Lumbricus* earthworm and SOCS2.

### 5.4. Protein–Protein Docking Simulation

In this section, we thoroughly investigated the interactions between the proteins and peptides obtained from *Lumbricus* and the SOCS2 target receptor. Employing protein–protein docking simulations, we aimed to uncover critical aspects of the binding mechanism. This included identifying pivotal residues crucial for forming protein–protein complexes, understanding the types of intermolecular interactions at play, assessing binding affinities, examining binding modes, and analyzing orientations. To delineate the binding sites of the target receptor, we utilized PDBSum, a computational tool tailored for detailed summaries of protein structures and their interactions [64]. This examination provided us with insights into how essential residues are spatially arranged within SOCS2′s binding site and their potential interactions with *Lumbricus*-derived proteins and peptides. To maintain the precision of our analysis, we refined the receptor using Swiss-PdbViewer v4.1.1 [65] before proceeding with protein–protein docking analysis, ensuring a reliable foundation for our subsequent investigations. To delve deeper into understanding whether the proteins and peptides derived from *Lumbricus* earthworms exhibit agonistic or antagonistic effects toward SOCS2, we utilized well-established molecules for comparison. EpoR peptide, identified by its PDB ID, 6I5N [63], serves as a standard agonist known to bind to SOCS2 within the PDB complex. In contrast, N4BP1, with a PDB ID of 8T48 [66], acts as a potent suppressor of E3 ligase activity. By comparing *Lumbricus*-derived compounds to these known standards, we aimed to evaluate their interactions with SOCS2, particularly focusing on binding pocket residues (Arg73, Ser75, Ser76, and Arg96) [67]. These residues are pivotal in determining the specificity and strength of interactions between SOCS2 and its binding partners. Following this, protein–protein docking calculations were conducted using the advanced interface option within the standalone version of High Ambiguity Driven Protein–protein Docking (HADDOCK). HADDOCK is a well-established computational tool for modeling protein–protein interactions, utilizing ambiguous interaction restraints derived from experimental or computational sources [68,69]. Using this method, we evaluated the interactions between *Lumbricus* proteins and peptides and SOCS2, predicting potential binding modes and affinities. Optimal protein–protein docking results for each complex were selected based on two key criteria: the highest number of observed clusters or populations, indicating the reliability of predicted interactions, and the highest docking score (HADDOCK score), which measures the strength of the binding affinity between *Lumbricus* proteins and peptides and SOCS2 within the protein–protein complex.

### 5.5. Molecular Dynamics (MD) Simulation

Molecular dynamics (MD) simulations were used to study the dynamics and stability of the protein–protein complexes involving *Lumbricus* proteins and peptides with SOCS2. These simulations were performed using GROMACS 2022.5, a well-established molecular dynamics simulation software renowned for its efficiency and accuracy in modeling biomolecular systems [70]. For the simulations, we utilized the Optimized Potentials for Liquid Simulations (OPLS-AA/L) force field, which accurately models molecular interactions within the system [71]. The dimensions of the simulation box were set using default cubic box parameters to accommodate the biomolecular complexes effectively. Standard protocols were followed to prepare simulation input files, which included adding water molecules using the Single Point Charge Extended (SPCE) model and incorporating counterions to maintain system neutrality [72]. Energy minimization was performed using the steepest-descent method to eliminate steric clashes and achieve system relaxation to a stable state. Following this, a two-phase equilibration process was executed. Initially, in phase 1, the system underwent equilibration in the NVT ensemble to regulate temperature fluctuations and stabilize conditions. Subsequently, in phase 2, equilibration was conducted in the NPT ensemble to maintain constant pressure and temperature. Once equilibrated, production MD simulations were conducted over 100 nanoseconds to observe the dynamics of the protein–protein complexes over an extended period. Throughout the simulations, parameters such as the root mean square deviation (RMSD), the root mean square fluctuation (RMSF), the radius of gyration (RoG), potential energies, and intermolecular hydrogen bonding interactions were monitored and analyzed to assess complex stability and conformational dynamics. The visualization of critical residues and intermolecular interactions within the predicted protein–protein complexes was conducted through manual inspection using molecular visualization software like PyMOL [73] and UCSF Chimera [74]. These tools enabled the examination and interpretation of essential structural characteristics and interactions within the simulated complexes, offering valuable insights into the mechanisms that govern the binding and stability of the proteins and peptides from *Lumbricus* with SOCS2.

### 5.6. Molecular Mechanics/Poisson–Boltzmann Surface Area (MM/PBSA) Calculations

The Molecular Mechanics/Poisson–Boltzmann Surface Area (MM/PBSA) method, employing MD simulations, was employed to investigate the protein–protein interactions involving proteins and peptides derived from *Lumbricus* and SOCS2. MD simulations produced a range of protein conformations, and representative snapshots were chosen for detailed analysis [75]. Each snapshot was subjected to thorough energy computations, encompassing calculations of gas-phase energy, the estimation of solvation energy using a continuum solvent model, and entropy calculations. These energy components were integrated to determine the binding free energy of the protein–protein complex [76,77]. To conduct these computations, we utilized the gmx_MMPBSA module available within the GROMACS simulation package [78,79]. This module facilitates the precise and efficient calculation of binding free energies for biomolecular complexes. The MM/PBSA method is renowned for its ability to predict the binding free energy of protein–protein interactions, establishing it as a valuable tool for understanding the energetic aspects of biomolecular interactions [80]. The MM/PBSA binding free energy calculation is derived from the following equation:Δ*G_binding* = Δ*G_complex* − Δ*G_proteinX*_1_ − Δ*G_proteinX*_2_
where the variables are defined as follows:

Δ*G_binding*: the binding free energy associated with the formation of the protein–protein complex.

Δ*G_complex*: the free energy of the fully solvated protein–protein complex.

Δ*G_proteinX*_1_: the free energy of protein 1 in its solvated state when unbound.

Δ*G_proteinX*_2_: the free energy of protein 2 in its solvated state when unbound.

The binding free energy was determined by calculating the difference between the free energy of the complex and the combined free energies of the unbound proteins. This calculation provided insights into the energetic alterations that occurred during the formation of the protein–protein complex, thereby elucidating the strength and stability of the interaction.

### 5.7. Statistical Analysis

In this study, statistical analysis was performed to investigate the relationships among all parameters derived from both molecular docking and molecular dynamics simulations. The data from these computational experiments underwent thorough statistical analysis and interpretation using SPSS 25 statistical software [81] and OriginLab Pro 2022 [82]. This analytical approach allowed for exploring correlations, trends, and patterns within the dataset, offering valuable insights into the relationships between various variables and parameters. Moreover, statistical analysis helped validate computational outcomes and identify significant findings, enhancing the understanding of the molecular interactions under scrutiny.

## 6. Conclusions

In conclusion, this study conducted a thorough investigation into the interactions between *Lumbricus*-derived proteins and SOCS2, revealing their potential therapeutic implications for cardiovascular diseases. Employing a comprehensive approach integrating bioinformatics, computational modeling, and molecular dynamics simulations, this research yielded valuable insights into the structural characteristics and binding affinities of *Lumbricus*-derived proteins and peptides with SOCS2. The findings indicated that certain proteins such as Lumbricin, Chemoattractive glycoprotein ES20, and Lumbrokinase-7T1 demonstrated activity comparable to standard antagonists in regulating SOCS2 function, suggesting their candidacy for therapeutic applications in cardiovascular diseases. Moreover, this study underscored the significance of computational methodologies in drug discovery, particularly for natural products possessing diverse chemical structures and biological activities. By leveraging advanced computational tools, this study efficiently screened a wide array of candidate compounds for their potential interactions with SOCS2, overcoming the challenges associated with traditional drug discovery methods. This approach not only accelerated the discovery process but also provided mechanistic insights into how these compounds operate at the molecular level. However, it is essential to recognize this study’s limitations, primarily its reliance on computational predictions that necessitate validation through experimental assays. Future research efforts should focus on validating these findings using biochemical and structural biology techniques, as well as conducting preclinical studies in relevant disease models. Furthermore, exploring the broader therapeutic potential of *Lumbricus*-derived proteins beyond cardiovascular diseases and investigating their synergistic effects with existing pharmacological agents could open new avenues for drug development and personalized medicine initiatives.

## Figures and Tables

**Figure 1 ijms-25-10818-f001:**
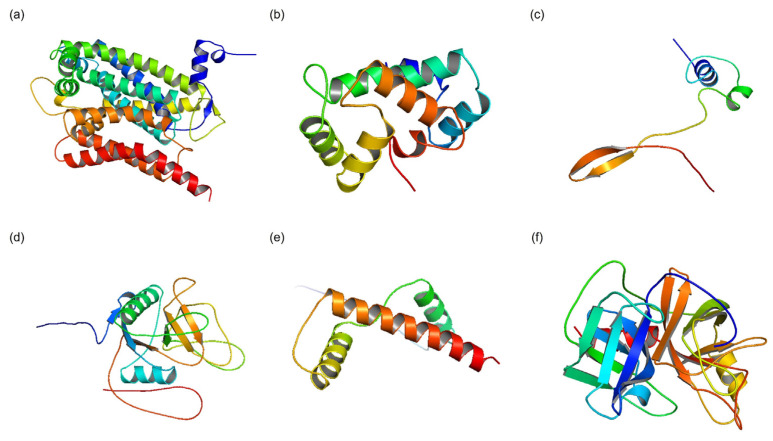
Three-dimensional structural modeling results illustrating proteins and peptides derived from earthworms of the *Lumbricus* genus. (**a**) Cytochrome b. (**b**) SCBP3 protein. (**c**) Lumbricin. (**d**) Chemoattractive glycoprotein ES20. (**e**) Histone H3. (**f**) Lumbrokinase-7T1. Three proteins exhibit well-folded, globular structures, while another three display large, unfolded regions or loops, indicating potential structural fluctuations that may influence their interactions with SOCS2 during docking.

**Figure 2 ijms-25-10818-f002:**
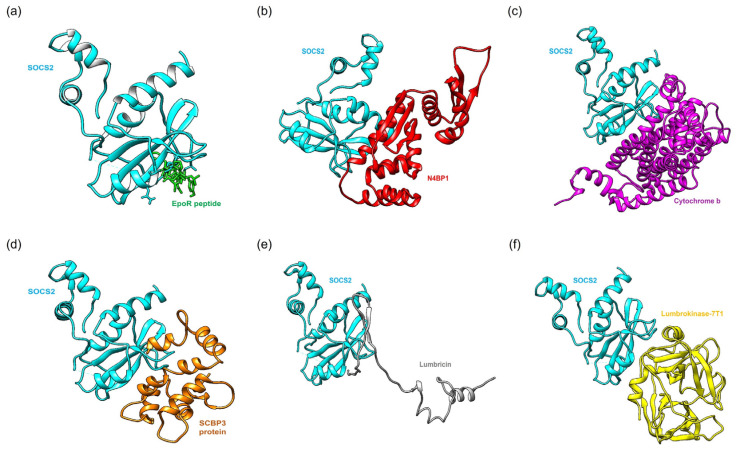
Molecular docking simulations illustrate the optimal binding orientations for interactions between proteins derived from the earthworm (*Lumbricus* genus) and SOCS2. (**a**) SOCS2: EpoR peptide (agonist) complex. (**b**) SOCS2: N4BP1 (antagonist) complex. (**c**) SOCS2: Cytochrome b complex. (**d**) SOCS2: SCBP3 protein complex. (**e**) SOCS2: Lumbricin complex. (**f**) SOCS2: Lumbrokinase-7T1 complex.

**Figure 3 ijms-25-10818-f003:**
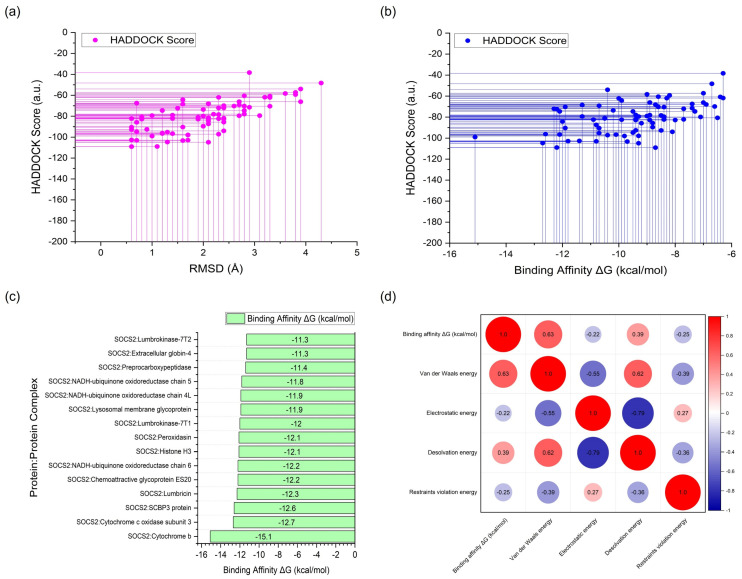
Summary of molecular docking outcomes. (**a**) Correlation between HADDOCK score and RMSD. (**b**) Correlation between HADDOCK score and binding affinity. (**c**) Binding affinity values of the top-performing proteins derived from *Lumbricus* earthworms, with a threshold of −11.0 kcal/mol. (**d**) Correlation matrix illustrating the relationship between binding energy (kcal/mol) and individual energy components.

**Figure 4 ijms-25-10818-f004:**
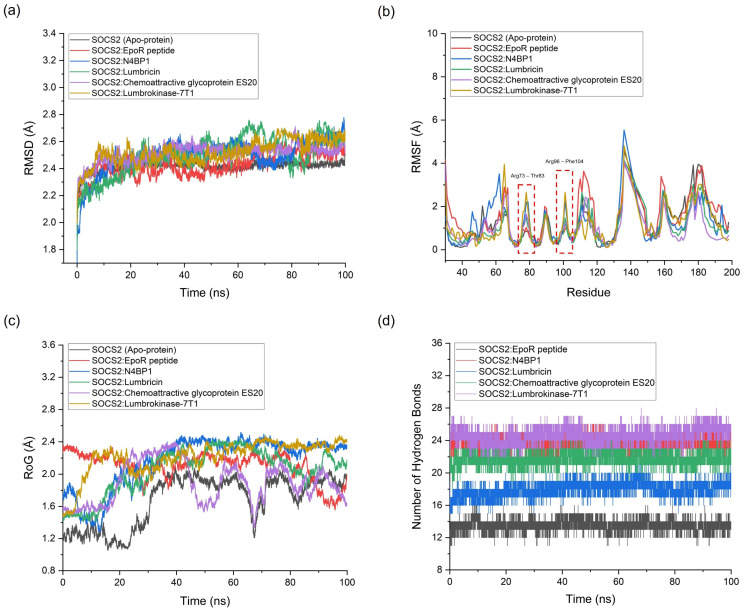
An analysis of molecular dynamics (MD) simulations for complexes formed between proteins sourced from the *Lumbricus* genus earthworm and SOCS2, including several key parameters: (**a**) the root mean square deviation (RMSD) assessed structural stability, (**b**) the root mean square fluctuation (RMSF) depicted residue flexibility, (**c**) the radius of gyration (RoG) illustrated structural compactness, and (**d**) the number of hydrogen bonds highlighted intermolecular interactions.

**Figure 5 ijms-25-10818-f005:**
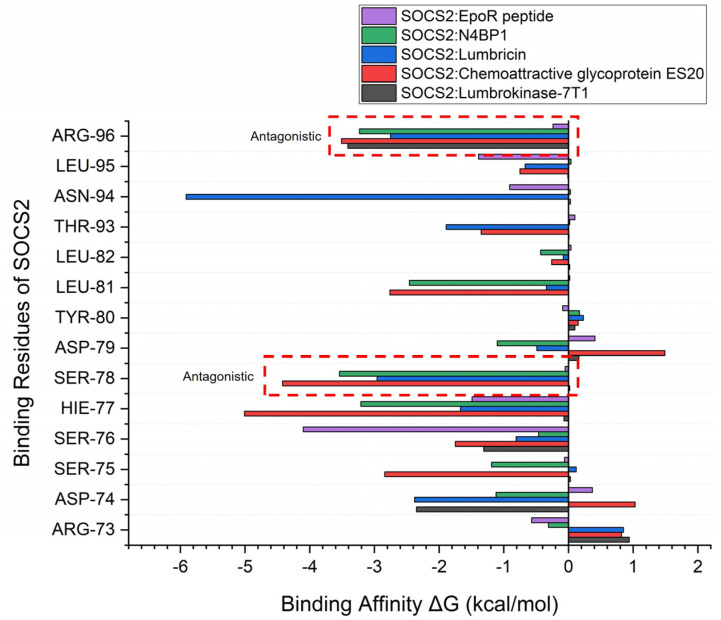
An analysis of the binding free energy for individual amino acids in SOCS2 interactions with the EpoR peptide (standard agonist), N4BP1 (standard antagonist), and the top three proteins derived from earthworms (*Lumbricus* genus) using MM/PBSA calculations.

**Table 1 ijms-25-10818-t001:** Selected proteins and peptides derived from *Lumbricus* earthworm.

Protein/Peptide	UniProt ID	Sequence	Size (kDa)
Actin-1	P92182	MCDEEVTALVVDNGSGMCKAGFAGDDAPRAVFPSIVGRPRHQGVMVGMGQKDSYVGDEAQSKRGILTLKYPIEHGIVTNWDDMEKIWHHTFYNELRVAPEEHPVLLTEAPLNPKANREKMTQIMFETFNSPAMYVAIQAVLSLYASGRTTGIVLDSGDGVTHTVPIYEGYALPHAILRLDLAGRDLTDYLMKILTERGYSFTTTAEREIVRDIKEKLCYVALDFDQEMGTAASSSSLEKSYELPDGQVITIGNERFRCPESMFQPAFLGMESAGIHETTFNSIMKCDVDIRKDLYANTVMSGGTTMFPGIADRMQKEITSMAPSTMKIKIIAPPERKYSVWIGGSILASLSTFQQMWISKQEYDESGPSIVHRKCF	41.85
Chemoattractive glycoprotein ES20	O44335	MKTYLLLVFLVGAHALVCPPGFTYLPAGESCYKVIFESHDWHSATERCRQESRGLAAISTPEESIAVKEFIDTEISKDSAGAAVCHPTGQSGIRFWTSGLQTKDTCTKTSFLLKITNTFEVPFDFTNWADGEPTLPRKTEKFSALIVGSSERTPSGTTMTATSSCVHSANISNDTLKRISVLPHLYVGFCDEIWLYLNFCLISIQILI	22.96
Cytochrome b	Q34945	MFKPIRTTHPAIKIINSTLIDLPAPNNISIWWNYGSLLGLCLVIQVLTGLFLSMHYVPNIEMAFSSVALISRDVNYGWLLRSIHANGASMFFLFIYLHAGRGLYYGSYNLSETWNIGVILFLLTMATAFMGYVLPWGQMSFWGATVITNLFSAIPYIGKTLVEWIWGGFAVDNATLNRFFAFHFILPFAIMGATILHIMFLHESGSNNPIGLNADSDRIPFHPYYSIKDTLGYTLAISALSLMVLFEPNLFTDPENFLMANPLVTPIHIKPEWYFLWMYAILRSIPNKLGGVMALFAAIVILFIPPLTSVMNKRSLSFYPLNKTMFWGLVASWAILTWIGGRPVEDPFIIIGQVFTSLYFIYFISSPTISKLWDDSIII	42.88
Fibrinolytic enzyme	P83298	VIGGTNASPGEFPWQLSQQRQSGSWSHSCGASLLSSTSALSASHCVDGVLPNNIRVIAGLWQQSDTSGTQTANVDSYTMHENYGAGTASYSNDIAILHLATSISLGGNIQAAVLPANNNNDYAGTTCVISGWGRTDGTNNLPDILQKSSIPVITTAQCTAAMVGVGGANIWDNHICVQDPAGNTGACNGDSGGPLNCPDGGTRVVGVTSWVVSSGLGTCLPDYPSVYTRVSAYLGWIGDNSR	24.84
Histone H3	A0A1C9UP21	GGKAPRKQLATKAARKSAPATGGVKKPHRYRPGTVALREIRRYQKSTELLIRKLPFQRLVREIAQDFKTDLRFQSSAVMALQEASEAYLVGLFEDTNLCAIH	11.47
Lumbricin	O96447	MSLCISDYLYLTLTFSKYERQKDKRPYSERKNQYTGPQFLYPPERIPPQKVIKWNEEGLPIYEIPGEGGHAEPAAA	8.85
Lumbrokinase-7T1	B8ZZ01	MRSFVAFLAALSLCQARPQKFLDGARPSFRMGGEQYIIGGSNASPGEFPWQLSQTRGGSHSCGASLLNALNGLSAFHCVDGAAPGTITVIAGLHDRSGTPGSQEVDITGYTMHENYNQGTNTYANDIAILHFASAINIGGNGQAALLPANNDNDYSGLTCVISGWGRKGSSNVLPDTLQKASIQVIGTTQCQSLMGSIGHIWDNHICLYNNTNNVGSCNGDSGGPLNCPDGGTRVAGVTSWGVSSGAGNCLQTYPTVYTRTSAYLSWIANNS	28.35
Ribosomal protein S27	Q9U5N5	MPLTRDLLHPTLKDEKRKCKLKRLVQSPNSFFMDVKCPGCYKITTVFSHAQTVVLCVGCNTVLCQPTGGKARLTEGCSFRRKQH	9.50
SCBP3 protein	Q7YWL4	VWEQYLKGVVSDGTRLTQAVFVEAVKKQLGDPNFKKVLAGPLPLFFSAVDGNGDGLIQKDEFQLFFKLLGIPESAEKSFEAIDTNKDGDISKEEFVIAGTDFFTSTDESSPSKYFWGPLV	13.25
Ubiquitin	P84589	TITLEVEPSDTIENVKAKIQDKEGIPPDQQRLIFAGKQLEDGRTLSDYNIQKESTLHLVLRLR	7.20

**Table 2 ijms-25-10818-t002:** Molecular docking outcomes: binding affinities and structural attributes of SOCS2 interactions with proteins and peptides obtained from *Lumbricus* earthworms.

Complex	HADDOCK Score (a.u.)	Binding Affinity ΔG (kcal/mol)	Kd (nM)	Cluster Size	RMSD (Å)
Standard					
SOCS2: EpoR peptide (standard agonist)	−81.0 ± 3.5	−8.8	650	22	1.7 ± 0.1
SOCS2: N4BP1 (standard antagonist)	−94.3 ± 11.4	−8.3	1500	17	1.2 ± 0.3
Protein and peptide derived from earthworm (*Lumbricus* genus)					
SOCS2: Cytochrome b	−99.0 ± 8.2	−15.1	0.02	27	1.0 ± 0.6
SOCS2: Cytochrome c oxidase subunit 3	−104.7 ± 6.3	−12.7	1.09	15	1.3 ± 0.5
SOCS2: SCBP3 protein	−96.2 ± 6.2	−12.6	1.30	41	1.2 ± 0.9
SOCS2: Lumbricin	−72.1 ± 3.7	−12.3	2.10	43	2.3 ± 0.3
SOCS2: Chemoattractive glycoprotein ES20	−72.4 ± 5.3	−12.2	2.30	20	1.5 ± 1.0
SOCS2: NADH-ubiquinone oxidoreductase chain 6	−109.0 ± 7.1	−12.2	2.30	14	1.1 ± 0.7
SOCS2: Histone H3	−96.6 ± 4.0	−12.1	3.09	8	1.2 ± 0.2
SOCS2: Peroxidasin	−74.6 ± 7.6	−12.1	2.70	8	2.8 ± 0.0
SOCS2: Lumbrokinase-7T1	−84.3 ± 2.8	−12.0	3.69	43	2.1 ± 0.2
SOCS2: Lysosomal membrane glycoprotein	−70.3 ± 2.3	−11.9	4.29	5	2.6 ± 0.9
SOCS2: NADH-ubiquinone oxidoreductase chain 4L	−90.4 ± 5.4	−11.9	4.10	12	1.6 ± 1.0
SOCS2: NADH-ubiquinone oxidoreductase chain 5	−102.9 ± 11.6	−11.8	4.69	7	1.7 ± 0.3
SOCS2: Preprocarboxypeptidase	−102.7 ± 9.1	−11.4	9.70	19	0.6 ± 0.4
SOCS2: Extracellular globin-4	−79.4 ± 10.8	−11.3	12.00	6	1.0 ± 0.6
SOCS2: Lumbrokinase-7T2	−68.5 ± 2.5	−11.3	10.99	35	1.6 ± 0.5

**Table 3 ijms-25-10818-t003:** Intermolecular contacts and non-interacting surface areas for SOCS2 complexes with standard agonist, antagonist, and earthworm-derived proteins and peptides.

Complex	ICs Charged-Charged	ICs Charged-Polar	ICs Charged-Apolar	ICs Polar-Polar	ICs Polar-Apolar	ICs Apolar-Apolar	NIS Charged	NIS Apolar
Standard								
SOCS2: EpoR peptide (standard agonist)	3	8	12	2	11	12	25.55	38.69
SOCS2: N4BP1 (standard antagonist)	5	11	20	6	11	8	29.71	39.49
Protein and peptide derived from earthworm (*Lumbricus* genus)								
SOCS2: Cytochrome b	0	0	25	0	38	33	15.13	52.85
SOCS2: Cytochrome c oxidase subunit 3	2	7	22	6	30	26	14.71	49.41
SOCS2: SCBP3 protein	2	3	24	2	27	16	27.83	41.74
SOCS2: Lumbricin	7	9	20	0	22	12	26.34	40.98
SOCS2: Chemoattractive glycoprotein ES20	8	15	24	5	23	16	22.98	42.39
SOCS2: NADH-ubiquinone oxidoreductase chain 6	0	3	21	7	32	29	14.29	52.01
SOCS2: Histone H3	7	6	27	0	20	14	27.56	42.67
SOCS2: Peroxidasin	13	20	24	5	22	6	26.23	41.86
SOCS2: Lumbrokinase-7T1	3	15	12	4	24	10	17.72	41.14
SOCS2: Lysosomal membrane glycoprotein	12	17	18	5	18	11	20.49	39.02
SOCS2: NADH-ubiquinone oxidoreductase chain 4L	0	8	21	2	24	14	17.73	46.82
SOCS2: NADH-ubiquinone oxidoreductase chain 5	0	3	17	1	25	29	14.12	50.10
SOCS2: Preprocarboxypeptidase	7	17	23	5	20	28	26.18	40.05
SOCS2: Extracellular globin-4	15	18	28	2	12	6	31.28	37.04
SOCS2: Lumbrokinase-7T2	5	10	16	9	24	11	19.03	41.69

Note: ICs: Number of intermolecular contacts; NIS: Non-interacting surface.

**Table 4 ijms-25-10818-t004:** A detailed examination of hydrogen bond interactions occurring between SOCS2 at its binding sites and the most effective proteins originating from earthworms (*Lumbricus* genus), focusing on identifying specific residues and atoms crucial for the binding mechanism.

Complex	Residue (Receptor)	Protein Atom (Receptor)	Residue (Interacting Protein/Peptide)	Protein Atom (Interacting Protein/Peptide)	Interaction Distance (Å)
SOCS2: EpoR peptide (standard agonist)	Val55	N	Asp8	OD1	2.76
	Ser76	OG	Asp8	OD1	3.09
	Arg96	NH1	Glu3	O	2.69
	Arg96	NH2	Glu3	O	3.33
	Lys113	NZ	Ser1	OG	2.73
SOCS2: N4BP1 (standard antagonist)	Gln32	NE2	Glu178	OE1	2.77
	Arg41	NH1	Glu118	OE1	2.65
	Arg41	NH2	Glu118	OE2	2.57
	Tyr49	OH	Lys132	NZ	2.78
	Asp74	OD2	Lys132	NZ	2.55
	Ser75	O	Asn136	ND2	3.04
	Ser78	O	Ser174	OG	2.63
	Asp79	OD2	Lys145	NZ	2.62
	Arg96	NH2	Glu138	OE1	2.67
SOCS2: Cytochrome b	His77	NE2	Leu122	O	2.74
	Arg96	NH2	Ile119	O	2.72
SOCS2: SCBP3 protein	Lys59	NZ	Asp50	O	2.58
	Arg96	NH1	Leu68	O	2.66
SOCS2: Lumbricin	Arg41	NH1	Glu56	O	3.08
	Arg41	NH1	Glu56	OE1	2.60
	Arg41	NH2	Glu56	O	2.98
	Tyr49	OH	Lys53	NZ	2.83
	Ser52	OG	Glu63	OE1	2.59
	Asp74	OD2	Lys53	NZ	2.65
	Ser78	OG	Ile46	O	2.67
	Asp79	OD1	Arg45	NH1	2.65
	Thr93	OG1	Glu72	OE2	2.70
	Asn94	N	Glu72	OE2	2.73
	Asn94	ND2	Glu72	OE1	2.64
	Arg96	NE	Glu72	O	2.74
	Arg96	NH2	Glu72	O	2.87
SOCS2: Lumbrokinase-7T1	Glu57	OE2	Lys20	NZ	2.55
	Asp74	OD2	Arg17	NH1	2.69
	Asp74	OD2	Arg17	NH2	2.72
	Ser76	O	Gln19	NE2	3.20
	Asp101	OD1	Asn249	ND2	2.66
	Asp101	OD2	Ser170	OG	2.66
	Cys111	O	Lys168	NZ	2.91
	Lys113	NZ	Gly32	O	2.87
	Lys113	NZ	Gln35	OE1	2.66
	Leu116	O	Asn214	ND2	2.98

**Table 5 ijms-25-10818-t005:** Time-averaged structural properties obtained from the MD simulations of SOCS2 protein–protein complexes.

Complex	Average RMSD (Å)	Average RMSF (Å)	Average RoG (Å)	Number of Hydrogen Bonds between the Two Proteins	Potential Energy (kcal/mol)
Standard					
SOCS2 (apo-protein)	2.417	1.089	1.674	N/A	−158,603.87
SOCS2: EpoR peptide (standard agonist)	2.423	1.397	2.101	11	−159,708.72
SOCS2: N4BP1 (standard antagonist)	2.496	1.179	2.162	24	−459,214.66
Protein and peptide derived from earthworm (*Lumbricus* genus)					
SOCS2: Cytochrome b	2.467	0.876	2.182	39	−471,304.36
SOCS2: Cytochrome c oxidase subunit 3	2.504	1.122	2.176	30	−450,869.63
SOCS2: SCBP3 protein	2.587	1.301	2.198	22	−316,369.57
SOCS2: Lumbricin	2.495	1.031	2.148	18	−694,628.89
SOCS2: Chemoattractive glycoprotein ES20	2.512	0.941	2.176	22	−621,068.14
SOCS2: NADH-ubiquinone oxidoreductase chain 6	2.487	1.001	2.178	20	−466,577.25
SOCS2: Histone H3	2.413	0.917	2.287	16	−629,402.13
SOCS2: Peroxidasin	2.599	1.106	2.678	35	−2,208,424.10
SOCS2: Lumbrokinase-7T1	2.523	1.123	2.213	27	−1,065,029.39
SOCS2: Lysosomal membrane glycoprotein	2.511	1.115	2.298	31	−1,645,741.31
SOCS2: NADH-ubiquinone oxidoreductase chain 4L	2.498	1.178	2.199	26	−406,049.87
SOCS2: NADH-ubiquinone oxidoreductase chain 5	2.487	1.101	2.190	21	−1,187,770.16
SOCS2: Preprocarboxypeptidase	2.596	1.259	2.188	30	−560,689.83
SOCS2: Extracellular globin-4	2.543	1.028	2.287	23	−473,808.36
SOCS2: Lumbrokinase-7T2	2.524	1.060	2.214	24	−764,436.84

**Table 6 ijms-25-10818-t006:** The average binding free energy (ΔG_binding_) for the SOCS2 protein–protein complexes is reported with standard deviation in kcal/mol units, as determined by MM/PBSA calculations.

Complex	MM/PBSA Calculation Results ΔG_binding_ (kcal/mol)			Average (kcal/mol)
	I	II	III	
Standard				
SOCS2: EpoR peptide (standard agonist)	−42.84	−43.48	−41.48	−42.60
SOCS2: N4BP1 (standard antagonist)	−42.34	−42.34	−42.85	−42.51
Protein and peptide derived from earthworm (*Lumbricus* genus)				
SOCS2: Cytochrome b	−50.57	−49.93	−49.82	−50.11
SOCS2: Cytochrome c oxidase subunit 3	−44.93	−44.93	−44.87	−44.91
SOCS2: SCBP3 protein	−29.33	−29.45	−29.45	−29.41
SOCS2: Lumbricin	−59.22	−59.26	−59.26	−59.25
SOCS2: Chemoattractive glycoprotein ES20	−53.69	−57.76	−53.60	−55.02
SOCS2: NADH-ubiquinone oxidoreductase chain 6	−52.81	−51.48	−53.15	−52.48
SOCS2: Histone H3	−44.05	−44.84	−45.82	−44.90
SOCS2: Peroxidasin	−42.91	−42.91	−42.91	−42.91
SOCS2: Lumbrokinase-7T1	−69.22	−69.35	−69.27	−69.28
SOCS2: Lysosomal membrane glycoprotein	−34.42	−34.89	−34.75	−34.69
SOCS2: NADH-ubiquinone oxidoreductase chain 4L	−48.14	−48.35	−48.35	−48.28
SOCS2: NADH-ubiquinone oxidoreductase chain 5	−37.74	−36.11	−36.11	−36.65
SOCS2: Preprocarboxypeptidase	−39.55	−38.03	−39.58	−39.05
SOCS2: Extracellular globin-4	−50.09	−49.68	−50.30	−50.02
SOCS2: Lumbrokinase-7T2	−46.18	−45.80	−47.11	−46.36

## Data Availability

The original contributions presented in this study are included in the article/Appendix A, and further inquiries can be directed to the corresponding author.

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
