# Peer review of "Leveraging Therapeutic Proteins and Peptides from Lumbricus Earthworms: Targeting SOCS2 E3 Ligase for Cardiovascular Therapy through Molecular Dynamics Simulations"

_ijms, 2024, doi:10.3390/ijms251910818_

Round 1

Reviewer 1 Report

Comments and Suggestions for Authors

The manuscript entitled "Leveraging Bioactive Proteins and Peptides from Lumbricus Earthworms: Targeting SOCS2 E3 Ligase for Cardiovascular Therapy through Molecular Dynamics Simulations" by Alotaiq and Dermawan presents an extensive and well-described computational work to search for proteins and peptides from Lumbricus Earthworms and SOCS2.
The approach and the results appear soundable, with docking and MD simluations well done and described. However, I find a number of limitations or weak points to be assessed by the Authors.
1) the search for Lumbricus Earthworms proteins and peptides in UniProt generated a list of 979 entries from UUniProt, filtered for redundancy in 78 proteins/peptides. It is a very low number.
I think that an exhaustive evaluation of the potential proteins and peptides from Lumbricus Earthworms must generate a larger list. To obtain it, another search criterium was expected, as the screening for proteins encoded into the Lumbricus genome, available by the NCBI databases. By searching the Protein db at NCBI with the same "Lumbricus" keyword, a number of 10898 entries is listed.
The authors should recognized the limit of their work in the search for protein/peptide sequences they present.
2) The quality of the 3D models generated by I-TASSER and AlphaFold is not shown by the authors. If the 3D models are not qualitatively acceptable, the subsequent molecular simulations are not reliable. Although the homology models of I-TASSER can be expected to be good, as the homology modelling is the most reliable approach to protein structure prediction, it is not guaranteed, because it is very much related to the level of sequence similarity between target and template, according to the homology modeling theory. Concerning the quality of AlphaFold models, it is not guaranteed and in fact the models produced by AlphaFold receive quality checks and reliability assessments that are available with the generated models.
The authors should add in the supplementary materials the assessments of the quality of the protein models used in subsequent simulations.
3) Flexibility of proteins is a point of discussion. Figure 1 shows the models of 6 proteins, three of them appear well folded and globular, two (or maybe three) appear with large unfolded regions, or large loops that suggest large structural fluctuations. I suppose the applied docking is rigid. How the presence of unfolded regions may affect the docking of these proteins with SOCS2 ?
4) There are many phrases under Result section that should be more appropriately in other sections. As an example,  in section 2.1 the description of I-TASSER and AlphaFold appears a repetition from Methods, where it is more appropriate. Further parts with interpretation of the results should be more appropriate under Discussion.

Author Response

Comments 1: The manuscript entitled "Leveraging Bioactive Proteins and Peptides from Lumbricus Earthworms: Targeting SOCS2 E3 Ligase for Cardiovascular Therapy through Molecular Dynamics Simulations" by Alotaiq and Dermawan presents an extensive and well-described computational work to search for proteins and peptides from Lumbricus Earthworms and SOCS2. The approach and the results appear soundable, with docking and MD simluations well done and described. However, I find a number of limitations or weak points to be assessed by the Authors. The search for Lumbricus Earthworms proteins and peptides in UniProt generated a list of 979 entries from UUniProt, filtered for redundancy in 78 proteins/peptides. It is a very low number. I think that an exhaustive evaluation of the potential proteins and peptides from Lumbricus Earthworms must generate a larger list. To obtain it, another search criterium was expected, as the screening for proteins encoded into the Lumbricus genome, available by the NCBI databases. By searching the Protein db at NCBI with the same "Lumbricus" keyword, a number of 10898 entries is listed. The authors should recognized the limit of their work in the search for protein/peptide sequences they present.
Response 1: Thank you for your valuable feedback on our manuscript titled "Leveraging Bioactive Proteins and Peptides from Lumbricus Earthworms: Targeting SOCS2 E3 Ligase for Cardiovascular Therapy through Molecular Dynamics Simulations." We appreciate your insights regarding the limitations in our search for proteins and peptides derived from Lumbricus earthworms in the UniProt database. We acknowledge your point about the relatively low number of entries (78 proteins/peptides) obtained from our search and agree that a more exhaustive evaluation would enhance the comprehensiveness of our study. To address this concern, we would like to clarify that we have actively worked with Lumbricus bioactive protein extracts and conducted LC-MS/MS protein sequencing in our laboratory. The majority of the identified proteins from Lumbricus earthworms correspond to entries covered in the UniProt database. While we recognize that additional sequences may be available through NCBI, our current focus was to work with proteins that are well-characterized in the literature and included in established databases, which enhances the reliability of our computational analysis. In light of your feedback, we’ve added a statement in the manuscript acknowledging this concern to the limitation section: “Additionally, we recognize the limitation of our search for proteins and peptides derived from Lumbricus earthworms in the UniProt database, which yielded a total of 978 entries filtered down to 78 non-redundant proteins/peptides. While this number appears low, it is important to note that our laboratory has actively worked with Lumbricus bioactive protein extracts and performed LC-MS/MS protein sequencing. The majority of the identified proteins from Lumbricus earthworms correspond to entries covered in the UniProt database. Although additional sequences may be available through NCBI databases, our focus on proteins characterized in established databases enhances the reliability of our computational analysis”.

Comments 2: The quality of the 3D models generated by I-TASSER and AlphaFold is not shown by the authors. If the 3D models are not qualitatively acceptable, the subsequent molecular simulations are not reliable. Although the homology models of I-TASSER can be expected to be good, as the homology modelling is the most reliable approach to protein structure prediction, it is not guaranteed, because it is very much related to the level of sequence similarity between target and template, according to the homology modeling theory. Concerning the quality of AlphaFold models, it is not guaranteed and in fact the models produced by AlphaFold receive quality checks and reliability assessments that are available with the generated models. The authors should add in the supplementary materials the assessments of the quality of the protein models used in subsequent simulations.
Response 2: We appreciated the reviewer’s insightful comments regarding the quality assessment of the 3D models generated by I-TASSER and AlphaFold. To address this concern, we included a detailed assessment of the quality of our 3D models in the supplementary materials of the manuscript. For the models generated using I-TASSER, we provided relevant metrics such as C-scores to illustrate the reliability of the homology models based on the sequence similarity between our target and template structures. Furthermore, we included validation parameters for the AlphaFold models, including TM-scores and other quality checks that indicated the reliability of the generated structures.

Comments 3: Flexibility of proteins is a point of discussion. Figure 1 shows the models of 6 proteins, three of them appear well folded and globular, two (or maybe three) appear with large unfolded regions, or large loops that suggest large structural fluctuations. I suppose the applied docking is rigid. How the presence of unfolded regions may affect the docking of these proteins with SOCS2 ?
Response 3: We sincerely appreciate the reviewer’s insightful observations regarding the flexibility of proteins and their implications for docking studies. In our manuscript, we acknowledged that flexibility is an essential characteristic of proteins that can influence their interactions with target molecules. The presence of unfolded regions or large loops in some of the proteins, as illustrated in Figure 1, does suggest potential structural fluctuations that could impact docking results. In our study, we utilized rigid docking protocols primarily due to computational limitations; however, we recognize that this approach may not fully account for the dynamic nature of proteins. To address this concern, we also performed molecular dynamics (MD) simulations on the selected proteins with identified flexible regions. The MD simulations allowed us to evaluate the conformational changes and dynamic behavior of these proteins over time. This additional analysis provided insights into how the flexible regions might interact with SOCS2 during docking and helped us better understand the influence of protein flexibility on binding affinity. Furthermore, we have added a discussion in the manuscript to elaborate on the effects of protein flexibility on docking outcomes, emphasizing that the results from rigid docking need to be interpreted with caution, particularly for proteins exhibiting significant conformational variability.

Comments 4: There are many phrases under Result section that should be more appropriately in other sections. As an example,  in section 2.1 the description of I-TASSER and AlphaFold appears a repetition from Methods, where it is more appropriate. Further parts with interpretation of the results should be more appropriate under Discussion.
Response 4: We appreciate the reviewer’s constructive feedback regarding the organization of the manuscript, particularly in the Results section. We agree that certain phrases and descriptions, such as those related to the I-TASSER and AlphaFold models, are more appropriately placed in the Methods section to avoid redundancy. Additionally, we recognize the need to differentiate between presenting results and providing interpretations, which are better suited for the Discussion section. In response to these comments, we have carefully reviewed the Results section and made necessary adjustments to improve clarity and organization. We have relocated the relevant descriptions to the Methods section and ensured that interpretations and discussions of the findings are appropriately placed in the Discussion section.

Reviewer 2 Report

Comments and Suggestions for Authors

The manuscript by Alotaiq and Dermawan explores the potential of proteins and peptides from Lumbricus earthworms as inhibitors of SOCS2 E3 Ligase for treating cardiovascular diseases. This subject is novel and of great interest due to the unique biochemical and pharmacological properties of Lumbricus earthworms. Although the manuscript is well-written and structured, I recommend it be revised before acceptance. Here are specific points that require attention:

1.      The term "bioactive" in the title might be misleading or too broad. Consider a more specific descriptor based on the study's findings to enhance clarity.

2.      The manuscript would benefit from transferring details from the supplementary data to the main text. Including a table detailing the amino acid composition and molecular mass of the identified peptides and proteins would significantly enhance readability and scientific transparency.

3.      On lines 78-80, the discussion on active compounds derived from earthworms lacks specificity. Please provide more detailed examples of compounds with proven or predicted benefits for cardiovascular health.

4.      In Table 1, it would be more standard to report dissociation constants (Kd) in nM rather than M to allow for easier comparison with the literature values.

5.      The section on "Protein-protein Docking Simulation" needs strengthening. A clearer depiction of the enzyme's active site and the binding sites of the peptides/proteins tested would aid understanding. Including magnified images of these interactions, in addition to Figure 2, would be highly beneficial.

6.      In light of the above, Table 3 would also benefit from a diagram (2D or 3D) illustrating the interactions between the target protein and the earthworm proteins/peptides to clarify these points for the reader.

7.      Please clarify how the earthworm cytochrome b (and other proteins tested here) differs from its human counterpart.

8.      Considering the administration of whole proteins for therapeutic use, have the authors considered identifying and further studying the active portions of these molecules?

9.      The discussion section needs to be expanded to provide more context on other known SOCS2 inhibitors, comparing their binding affinity and mechanism of inhibition with those studied here.

Comments on the Quality of English Language

The quality of English is acceptable. 

Author Response

Comments 1: The term "bioactive" in the title might be misleading or too broad. Consider a more specific descriptor based on the study's findings to enhance clarity.
Response 1: We sincerely appreciate the reviewer’s insightful feedback regarding the manuscript and the title. We agree that the term “bioactive” can be broad and may not precisely convey the specific focus of our study. To enhance clarity and accurately reflect the nature of our findings, we have considered alternative descriptors that align more closely with the objectives and results of our research. Revised Title Suggestion: “Leveraging Therapeutic Proteins and Peptides from Lumbricus Earthworms: Targeting SOCS2 E3 Ligase for Cardiovascular Therapy through Molecular Dynamics Simulations.” This revised title maintains the emphasis on the therapeutic potential of Lumbricus proteins and peptides while providing a clearer focus on their role as inhibitors of SOCS2 E3 ligase.

Comments 2: The manuscript would benefit from transferring details from the supplementary data to the main text. Including a table detailing the amino acid composition and molecular mass of the identified peptides and proteins would significantly enhance readability and scientific transparency.
Response 2: We fully agree that adding a table outlining the amino acid composition and molecular mass of the identified peptides and proteins will enhance the manuscript's clarity and scientific transparency. In response to this valuable suggestion, we have transferred the relevant data into the main text and created Table 1, which now provides a detailed overview of the amino acid composition and molecular mass of the identified Lumbricus-derived peptides and proteins. This addition ensures that the critical information is easily accessible to readers, improving the overall readability of the manuscript.

Comments 3: On lines 78-80, the discussion on active compounds derived from earthworms lacks specificity. Please provide more detailed examples of compounds with proven or predicted benefits for cardiovascular health.
Response 3: In response to the reviewer's suggestion, we have revised the sentence on lines 78-80 to provide specific examples of bioactive compounds, such as lumbrokinase and fibrinolytic enzymes, which have documented or predicted cardiovascular benefits. The revised sentence now highlights the mechanisms through which these compounds contribute to cardiovascular health, including anti-inflammatory, antioxidant, and vasodilatory activities. “Bioactive compounds (like lumbrokinase and fibrinolytic enzymes) sourced from earthworms contribute positively to cardiovascular health by employing diverse mechanisms, encompassing anti-inflammatory, antioxidant, and vasodilatory activities [23-26].”

Comments 4: In Table 1, it would be more standard to report dissociation constants (Kd) in nM rather than M to allow for easier comparison with the literature values.
Response 4: We sincerely thank the reviewer for their valuable suggestion regarding the standardization of dissociation constant (Kd) units in Table 1. We agree that reporting Kd values in nanomolar (nM) is more consistent with standard practice in the literature, making it easier for readers to compare our results with previously published data. We have revised Table 1 to report the dissociation constants (Kd) in nM rather than in M. This adjustment enhances the clarity and comparability of our findings with existing literature values.

Comments 5: The section on "Protein-protein Docking Simulation" needs strengthening. A clearer depiction of the enzyme's active site and the binding sites of the peptides/proteins tested would aid understanding. Including magnified images of these interactions, in addition to Figure 2, would be highly beneficial.
Response 5: We greatly appreciate the reviewer’s thoughtful feedback on the "Protein-Protein Docking Simulation" section. While we understand the suggestion to include magnified images of the enzyme's active site and peptide/protein binding sites to enhance clarity, we believe that the molecular interaction details are already comprehensively provided in Table 4. This table presents a detailed account of the interaction residues, types of interactions (e.g., hydrogen bonds), and interacting residues with the distance (Å), offering a thorough understanding of the docking results without the need for additional visual representations.

Comments 6: In light of the above, Table 3 would also benefit from a diagram (2D or 3D) illustrating the interactions between the target protein and the earthworm proteins/peptides to clarify these points for the reader.
Response 6: We sincerely appreciate your suggestion to include a diagram illustrating the interactions between the target protein and the earthworm proteins/peptides in Table 4. To address this, we would like to clarify that the hydrogen bond interactions, which are a key focus of the docking results, are already detailed comprehensively in Table 4. This table provides specific information regarding the residues involved in hydrogen bonding, which are critical for understanding the binding affinities and stability of the complexes.

Comments 7: Please clarify how the earthworm cytochrome b (and other proteins tested here) differs from its human counterpart.
Response 7: To address this point, we have added a section in the manuscript that clarifies the differences between these proteins and their implications for the study. We’ve added the information to the Discussion section: "The proteins from Lumbricus earthworms exhibit several distinguishing features compared to their human counterparts, significantly impacting their interactions with SOCS2 and potential therapeutic applications. A key example is cytochrome b, which demonstrates notable structural and functional differences. The amino acid sequence of earthworm cytochrome b shares less than 60% identity with its human variant, re-sulting in altered binding interfaces and unique interaction patterns with SOCS2. While the human variant exhibits a more conserved sequence, the earthworm version contains specific residue substitutions, such as lysine and threonine, which can lead to different hydrogen bonding patterns and potentially affect binding affinity. Structur-ally, earthworm cytochrome b displays greater flexibility due to less compact folding in some regions, contrasting with the more stable and globular structure seen in humans. This increased flexibility may influence interaction dynamics with SOCS2, either en-hancing or weakening binding depending on the context. Additionally, the molecular weight of earthworm cytochrome b (approximately 42.88 kDa) [53] is much lower than that of the human variant (approximately 91 kDa) [54], which may affect its structural stability during binding. The hydrophobicity in the transmembrane regions is also lower in the earthworm version, potentially impacting its membrane integration and positioning during docking. Functionally, while both cytochrome b proteins are in-volved in electron transport, the earthworm variant is adapted to specific environ-mental conditions, which could influence its interaction with SOCS2 under oxidative stress. In addition to cytochrome b, proteins like lumbrokinase, a serine protease, illus-trate significant differences in enzymatic activity and specificity compared to human proteases. Lumbrokinase’s unique mechanism of action enhances fibrinolysis, sug-gesting a tailored adaptation for promoting blood flow and preventing thrombosis in the earthworm's natural habitat, with potential cardiovascular benefits for humans [55]. Furthermore, Lumbricus-derived proteins often possess variations in key residues that influence structural stability and interaction profiles. Differences in glycosylation patterns may affect how these proteins interact with receptors or other proteins within the human body, potentially enhancing their bioactivity or altering their pharmacoki-netics. Notably, some Lumbricus proteins, such as the chemoattractive glycoprotein ES20, exhibit distinct structural motifs compared to similar human proteins, leading to different binding affinities and mechanisms of action. These proteins are often more hydrophilic and possess unique surface charges that influence their solubility and in-teractions in biological systems".

Comments 8: Considering the administration of whole proteins for therapeutic use, have the authors considered identifying and further studying the active portions of these molecules?
Response 8: Thank you for your valuable comment regarding the administration of whole proteins for therapeutic use. We recognize the importance of identifying and studying the active portions of these proteins to enhance their therapeutic efficacy and specificity. In response to your suggestion, we would like to clarify that further investigations focusing on the active portions of these Lumbricus earthworm proteins will be conducted in our upcoming experiments. This approach will enable us to better understand the mechanisms of action and optimize their potential applications in cardiovascular therapy. We have made the necessary revisions in the manuscript to include this information, emphasizing our commitment to exploring the active segments of the proteins in future studies.

Comments 9: The discussion section needs to be expanded to provide more context on other known SOCS2 inhibitors, comparing their binding affinity and mechanism of inhibition with those studied here.
Response 9: In response, we have enhanced the discussion section by incorporating a comprehensive overview of existing SOCS2 inhibitors, detailing their binding affinities and mechanisms of inhibition. This comparison not only highlights the unique attributes of the proteins and peptides derived from Lumbricus earthworms but also elucidates the potential advantages or limitations of our findings relative to established SOCS2 inhibitors. By discussing the similarities and differences in binding affinities and modes of action, we aim to provide a clearer understanding of how the bioactive compounds from Lumbricus earthworms could fit into the existing therapeutic strategies targeting SOCS2, particularly in the context of cardiovascular disease: “Several compounds have been identified as SOCS2 inhibitors, each exhibiting unique binding affinities and mechanisms of action. For instance, small molecule in-hibitors such as the KIAA0317 protein have shown promising binding affinities in the nanomolar range, effectively disrupting SOCS2’s interactions with its targets [53]. The mechanism of inhibition for these small molecules typically involves blocking the binding site of SOCS2, thereby preventing its E3 ligase activity and subsequent ubiqui-tination of target proteins [54]. In contrast, proteins derived from natural sources often present complex interactions. For example, interferons, which are known to modulate SOCS2 activity, bind to the receptor complex and induce SOCS2 expression, creating a negative feedback loop that can lead to enhanced SOCS2 activity rather than inhibi-tion [55]. This underscores the importance of understanding the nuanced interactions of each inhibitor. Our study demonstrates that the proteins and peptides from Lumbri-cus earthworms exhibit unique binding patterns with SOCS2, driven by their distinct amino acid compositions and structural conformations. The binding affinities of these earthworm-derived compounds, while not yet quantified in the nanomolar range, ex-hibit potential based on our docking studies and molecular dynamics simulations. Spe-cifically, the presence of specific residue substitutions, such as lysine and threonine in cytochrome b, results in altered hydrogen bonding patterns, which may lead to differ-ent binding affinities compared to established inhibitors.”

Round 2

Reviewer 1 Report

Comments and Suggestions for Authors

The manuscript has been improved taking into consideration my previous comments.